# Tooth Position in Wind Instrument Players: Dentofacial Cephalometric Analysis

**DOI:** 10.3390/ijerph18084306

**Published:** 2021-04-19

**Authors:** Miguel Pais Clemente, André Moreira, Catarina Morais, José Manuel Amarante, Afonso Pinhão Ferreira, Joaquim Mendes

**Affiliations:** 1Departamento de Cirurgia e Fisiologia, Faculdade de Medicina, Universidade do Porto, 4099-319 Porto, Portugal; amarante@med.up.pt; 2INEGI, Laeta, Labiomep, 4200-465 Porto, Portugal; jgabriel@fe.up.pt; 3Faculdade de Medicina Dentária, Universidade do Porto, 4200-135 Porto, Portugal; andre.luis.sa.moreira@gmail.com (A.M.); catarinamoraisroque@outlook.com (C.M.); aferreira@fmd.up.pt (A.P.F.); 4Faculdade de Engenharia, Universidade do Porto, 4200-465 Porto, Portugal

**Keywords:** cephalometric analysis, embouchure, orthodontics, performing arts medicine, string instruments, tooth position, wind instruments

## Abstract

Background: Specific dentofacial characteristics in wind instrumentalists should be taken in consideration when analyzing physiological and anatomical issues regarding the musician’s embouchure, posture, and biomechanics during musical performance. Objectives: To compare tooth cephalometric characteristics between wind instrument players and string players (overjet, overbite, lower facial height, facial convexity, lower incisor inclination, and interincisal angle). Methods: In total, 48 wind instrumentalists (67%) and 24 string instrumentalists (33%). These musicians performed lateral tele-radiography and the correspondent linear and angular measurements of the dentofacial cephalometric analysis. Statistical comparison of wind and string instrumentalists was made by using an independent t-test. Results: Small variations on the analyzed parameters were found between the wind and string instrument groups. Based on the cephalometric analysis the variable interincisal angle was statistically significant (*p* < 0.05), when comparing the wind and string instrument group. Conclusions: Knowledge of the overjet and overbite value permits a substantial analysis on the tooth position of wind instrument players, where both of these parameters are increased and greater than the norm value. The cephalometry was an added value on the interpretation of possible factors that lead to the position of the central incisors of wind instruments. Till some extent in this group of musicians the applied forces during the embouchure mechanism on the anterior teeth and the existing perioral forces promote an equilibrium on the vector of forces. This study findings demonstrate that when evaluating the two samples, wind and string instruments there are different dentofacial configurations, however the only statistically significant differences that were found are related to the interincisal angle (*p* < 0.05).

## 1. Introduction

Performing arts medicine is a discipline involving professionals of different domains in the analysis, diagnosis, and treatment of health issues related to musical performers, painters, and dancers [1,2,3,4,5,6,7,8].

It was given the chance to understand the playing related topics, that can range from microbial contamination of musical wind instruments to somatosensory function, reflux symptoms, respiratory function, and ergonomics issues in wind instrumentalists [3,9,10,11]. Understanding certain professional’s dilemmas can bring to light eventual solutions that appear as occupational maladies, in an activity that is so demanding in terms of physical and psychological factors.

Within this perspective, it is notorious that a correct evaluation, characterization, and quantification of specific parameters of these individuals, the more the scientific community, namely the health professionals, will be able to comprehend, treat, and monitor the musician’s health along their career.

Playing related musculoskeletal disorders (PRMD) have been described [3,12,13] as the association between playing a musical instrument and the prevalence of temporomandibular disorders [14,15,16,17,18].

A special attention has been given to the stomatognathic system and wind instrument players [19,20,21] regarding the embouchure, more specifically to the orofacial structures involved in such precise mechanism that involve the jaws, the temporomandibular joint, the orofacial muscles, the tongue, the soft palate, the lips, and the teeth. It is true that the embouchure phenomenon starts in the respiratory system, where the lungs allow the musician to blow air towards the mouthpiece, independently of being a brass instrumentalist or a woodwind instrument player. However, the stabilization of the air flow, in the same way that is made by the diaphragmatic musculature, thoracic, cervical, and orofacial muscles, it is in the oral cavity where there is an intimate contact of the mouthpiece with the teeth.

This topic has been addressed, namely in terms of occlusion and craniofacial morphology [22,23,24]. Some investigations with wind instrumentalists have been carried out, using questionnaires, photographs, clinical examinations, and plaster models [25,26,27]. However, analyzing, specifically, the relationship of the anterior teeth of the upper jaw with the lower jaw, is somehow not so common in terms of cephalometric analysis, even though some attempts have been done in the past [28,29,30,31] since this is one interesting factor regarding a wind instrument player: the tooth position.

It is known that many wind instrumentalists mention that at the end of concerts or after many hours of rehearsals the teeth seem to have mobility. The applied forces during the embouchure of a particular instrument, such as the saxophone or the trumpet, can correspond to medium and heavy forces, like orthodontic forces, being transmitted to the orofacial structures [32]. Fortunately, these forces applied to the perioral structures are intermittent, however will they be sufficient to alter the tooth position? The determinants of the embouchure mechanism are different, even within the same group of instrument players like single reed instrumentalists, were the insertion of the mouthpiece inside the oral cavity is completely different in a clarinet player or a saxophonist—Are these aspects sufficient to change the tooth position in each particular group?

The aim of this study was to evaluate the tooth position of wind instrument players by comparing cephalometric values regarding dental parameters between two different groups of musicians: wind instrument players and string players. The parameters tested were overjet, overbite, lower facial height, facial convexity, lower incisor inclination, and interincisal angle.

## 2. Materials and Methods

This study involved 48 wind instrumentalists (67%) and 24 string instrumentalists (33%) from the Porto’s national orchestra, Casa da Música, and students from the Master of Science degree in Music and Performing Arts of Oporto (ESMAE), with an age comprehended between 18 to 40 years old, while the sex distribution of these musicians were the following, 32 women and 40 men. The inclusion criteria were adults (>18 years old), with more than 10 years of experience while playing their instrument as main instrument during musical training. The exclusion criteria were participants that had a prior orthodontic treatment or any musician that presented a history of maxillofacial surgery or mandibular injuries.

The present study was approved by the ethics committee of Faculty of Dental Medicine, University of Porto, no. 880292. Thus, it was in accordance with the World Medical Association Declaration of Helsinki. To all participants a verbal explanation was given together with a written consent explaining the objective of the study, its methods and risks and benefits.

To collect the lateral cephalograms (Figure 1) Orthoralix 9200- Gendex, KaVo, Biberach an der RiB, Germany, from the Faculdade de Medicina Dentária da Universidade do Porto was used. The images were taken to both string and wind instrumentalists, in maximum intercuspidation by the same technician to allow a standardization of the protocol. The participants were told to not move during the imaging acquisition, look forward, feet aligned, head in rest position, but in the orthostatic position. The participants were told to hold their head in a stabilized position with the olives in external auditory meatus, and with the indicator supporting the glabella. The subject’s sagittal plane was perpendicular to the path of the X-ray. The Frankfurt (horizontal) plane was parallel to the floor. To all participants was given a lead vest to minimize the radiation exposure.

For the cephalometric analysis, the same examiner confirmed all the values twice to ensure intra-examiner reliability. For this study it was considered the following parameters of the Rickett’s analysis:Interincisal angle: angle found between the upper incisal axis and lower incisal axis (in yellow, Figure 2a);Lower facial angle: angle between the planes formed by the anterior nasal spine to Xi point (ANS-Xi) and the Xi point to protuberance menti (Xi-PM) (in orange, Figure 2a);Facial convexity: direct measurement parallel to the Frankfort plane between point A and facial plane (N-Pog) (in purple, Figure 2b);Overjet: anterior-posterior overlap of the upper incisors over the lower incisors (in white, Figure 3a);Overbite: superior–inferior overlap of the upper incisors over the lower incisors measured relative to the incisal ridges (in blue, Figure 3a);Lower incisor protrusion: distance between the coronary extremity of the lower incisor to the A/Pog line (in red, Figure 3b).

The IBM SPSS Statistics version 24.0 (IBM Corp., Armonk, NY, USA) was used to obtain the variables distribution and posteriorly perform an independent t-test to compare the wind instrumentalists’ group with the string instrumentalists group searching for differences in the variable distribution. The null hypothesis (H0) for the present study was “wind and string instrumentalists have equal cephalometric parameters”. Thus, the alternative hypothesis (H1) stated that wind and string instrumentalists have differences regarding cephalometric parameters.

## 3. Results

For the variable interincisal angle the wind group showed a higher average comparatively to the string group, 127.46° and 122.34°, respectively. The trumpetists were the instrumentalists with the greater interincisal angle and the viola players with the smallest, 129.83° and 113.92°, respectively (Table 1). For the variable overjet the string group showed a higher average comparatively to the wind group, 4.55 and 4.05 mm, respectively. The French horn instrumentalists were the subgroup with the greater overjet and the bassoon with the smallest, 5.20 and 3.10 mm, respectively (Table 2). For the variable overbite the string group showed an higher average comparatively to the wind group, 3.24 and 3.05 mm, respectively. The transverse flute instrumentalists were the subgroup with the greater overbite and the saxophone with the smallest, 4.72 and 1.54 mm, respectively (Table 3). For the variable lower facial angle, the string group showed a lower average comparatively to the wind group, 42.10° and 43.63°, respectively. The bassoon instrumentalists showed the highest angle and the cello instrumentalists the smallest, 45.74 and 41.23°, respectively (Table 4). For the variable facial convexity, the string group showed a greater average comparatively to the wind group, 3.54 and 3.08 mm, respectively. The French horn instrumentalists showed the smaller value and the tuba the greatest value, 0.13 and 3.53 mm, respectively (Table 5). For the variable lower incisor protrusion, the string group showed a higher average comparatively to the wind group, 3.48 and 2.40 mm, respectively. The French horn instrumentalists showed the smallest value and the viola the highest, −0.367 and 5.60 mm, respectively (Table 6).

Comparing the wind and string groups by using an independent t-test, for all variables the null hypothesis was accepted, except for the interincisal angle (*p* < 0.05) (Table 7). Comparing the metal and woodwind groups by using an independent t-test, for all variables the null hypothesis was accepted (*p* < 0.05) (Table 8).

## 4. Discussion

The present study aimed to examine whether the tooth position could differ within musicians when playing different instruments. In this case, regarding the dentofacial cephalometric analysis between wind instrument players and string instrument players. The results of this study indicate that there are different dentofacial configurations, however the only statistically significant differences that were found were related to the interincisal angle (*p* < 0.05), when evaluating the two samples, wind instruments and string instruments. While comparing and analyzing the metal and the woodwind groups there was no parameter that was statistically significant, however there is a global consideration that should be taken regarding the fact that these musicians appeared with increased overjet and overbite values. This does not happen in regular patients, e.g., non-musicians. Usually, when there is an increased overjet there is a reduced overbite. With the inherent limitation that our study can represent in terms of the reduced number of participants inside the wind and the string instruments group, there will be an analysis to the descriptive statistics variables. This will be done along the discussion of each variable that the authors believe being relevant for the tooth position in particular of wind instrumentalists, regarding the influence of the mouthpiece on the orofacial region. The contact point of the wind instrumentalist mouthpiece can be an important factor for this interpretation and the results of the obtained tooth position in this current investigation.

Before a thorough cephalometric characterization of the study groups, in particular the wind instrumentalist group, it is relevant to contextualize the field of action of dental sciences regarding a sub-specialty that can be considered, performing arts medicine, when discussing the health and well-being issues of a musician.

Performing arts medicine addresses musculoskeletal and neuromuscular conditions that can be considered the main health topics related to instrumental musicians [1,3,4,6,7,8,13]. The musculoskeletal disorders are a sub-specialty where overuse syndrome, temporomandibular disorders, entrapment neuropathies, median neuropathy, radial neuropathy, cervical radiculopathy, focal dystonias, and joint hypermobility [1]. Therefore, it is important to implement preventive strategies where ergonomic and biomechanic considerations are part of the musician’s routine [1,33]. The implementation of health-related issues as a literacy procedure within these musicians should be a fundamental aspect to take in consideration, since the initiation of their musical activity. Relevant education and advice should be provided to musicians early in their injury whilst preventative information needs to be delivered early and throughout their careers [3].

Steinmetz et al. held a study with 84 musicians, with 93% showing dysfunctions in one or more of the examined postural stabilization systems (85% impairment of scapular stabilization system, 71% of lumbopelvic stabilization system, 57% upper crossed syndrome) [8]. Steintmetz et al. study suggests that insufficiencies of the postural stabilization systems play an important role in the manifestation of musculoskeletal pain and playing related musculoskeletal disorders (PRMD) in musicians [8]. The emphasis on the importance of physical examination where posture, range of motion, hypermobility, ergonomics is part of a physical examination is described by Jan Dommerholt [34]. Among many health care providers, the role of physiotherapists is highlighted by Jan Dommerholt, in the sense that these professionals are essential providers in the field of performing arts medicine, since they can play a substantial role in the prevention, diagnosis, and management of performance-related musculoskeletal injuries of musicians [35].

In a systematic review carried out by Christine Zaza, there were 7 eligible studies, PRMD point prevalence ranged from 39 to 87% in adult musicians, and from 34 to 62% in secondary school music students [12]. Available data indicate that the prevalence of PRMD in adult classical musicians is comparable to the prevalence of work-related musculoskeletal disorders reported for other occupational groups [12]. However, unlike workers in other occupations, musicians have no industry standards for occupational health and safety [12]. Christine Zaza refers that health care professionals’ awareness of the nature and extent of musicians’ health problems, as well as their awareness of treatment and information resources, has important clinical implications [12].

The biopsychosocial model created awareness about musician’s health; education on human anatomy and physiology in relation to playing the instrument; providing strategies for coping with anxiety, stress, and overcommitment; how to handle pain and discuss general health issues such as physical activity and nutrition [13]. Nevertheless, from the total of 170 randomized students of music conservatories in the Netherlands, it was possible to observe that the biopsychosocial prevention course tailored for musicians was not superior to physical activity promotion in reducing disability [13]. Independently to this was the advantage that performance-related disability and the presence of playing-related musculoskeletal disorders seem to decrease substantially in both groups over time [13].

Within this perspective it is interesting to notice that highlighting certain action components to musical students related with their body posture while playing is extremely relevant and from our point of view should focus a particular emphasis on the cranio-cervico-mandibular complex. Our experience when leading and treating musicians is that nowadays, musical teachers already start to give a particular attention to the musician’s occlusion and dental related aspects that can influence the student’s embouchure. The results of our study show specific dentofacial characteristics related to different woodwind and string instrumentalists that should be taken in consideration when analyzing physiological and anatomical issues regarding for example, the musician’s embouchure, posture and biomechanics during musical performance.

While certain postural defects appear to be common to all families of instruments, others are more characteristic of some families than others. The instrument associated with the best posture quality was the bagpipe, followed by percussion and strings [33]. Head alignment regarding the sagittal plane was in a forward position in 24 wind instrumentalists (*n* = 37), compared, for example, with the string instrumentalists, which had 8 presenting the same clinical condition (*n* = 13) [33]. The Blanco-Piñeiro et al. study says that vocalists, pianists, and the players of wind instruments tend to have relatively poor postures [33]. This can also bring to light the relevance of our study since the specific target of analyzing the tooth position of a wind instrumentalist can be indirectly associated to the body posture adopted by the musician. The weight and size of the instrument are supported mainly by the upper extremity of the human body, where the shoulder joint stabilizes the position of the forearm that presents a muscular activity with the triceps and extensor muscles maintaining the instrument in place, while the wrist as a synovial joint allows the hand of the musician and consequently the fingers to perform the necessary flexion or extension, making a desired pressure on the piston valve of the brass instruments, or the tone hole rings, for example, of a clarinet. The clarinet player undergoes the abduction of the 1st carpo-metacarpal joint, implying that the thumb of the right hand, usually develops a callus due to the pressure that is executed while holding the instrument. All these considerations were described from the anatomical site of the shoulder joint, where, of course, the cervical and thoracic muscles have a fundamental action on the stabilization of the cranio-cervico-mandibular complex. So, if a particular attention is made towards the proximal extremity of the wind instrument in relation to the head of the musician, it is notorious the major importance that the oral cavity has on the act of playing, allowing the musician to reach the notes of various harmonic series. For this to occur, as it was already mentioned there is a major importance of the pulmonary and respiratory function, to obtain the sound production while an air column passes through the anterior incisors and lips. These last anatomical landmarks, the teeth and lips are fundamental for the stabilization and equilibrium of the wind instrument mouthpiece during the embouchure mechanism, while the single and double reed instrumentalists have the vibration of the reed to produce sound and the brass instruments have the lips vibration. The induced pressures made by these different kind of wind instruments were previously monitored and quantified by Clemente et al., where it was possible to observe that certain instruments registered the following pressures at the perioral structures, the French horn (56–305 g), the transversal flute (220–305 g), and the trombone (201–325 g).

So, will these forces be sufficient to change the tooth position in wind instrument players?

A cross sectional observational study was carried out, comparing the occlusions of 170 professional musicians, subdivided according to type of instrument mouthpiece, and included 32 brass players with large cup-shaped mouthpieces, 42 brass players with small cup-shaped mouthpieces, and 37 woodwind players with single-reed mouthpieces [22]. A total of 59 string and percussion players formed the control group [19]. Impressions were taken of the teeth of each subject, and occlusal parameters were assessed from the study casts. No statistically significant differences were found in overjet, overbite, or crowding. Being possible to notice a significantly higher prevalence of lingual crossbites on the large-mouthpiece brass group in comparison with all other groups [17]. The conclusions of Grammatopoulos et al. was that playing a wind instrument does not significantly influence the position of the anterior teeth and is not a major etiologic factor in the development of a malocclusion, with exception to the predisposition that playing a brass instrument with a large cup-shaped can have on the development of lingual crossbites [31].

It is reported that long-term and repetitive playing of musical instruments, particularly stringed (violin and viola) and wind instruments can cause dysfunctions of the stomatognathic system [19]. These observations suggest that the problem should be mentioned, and orthodontic consultation should be proposed before starting to play an instrument [19]. Something that is in alignment with the purpose of our study is the evaluation of the tooth position in wind instrument players and also being able to evaluate the craniofacial morphology of a musician before the choice of a wind instrument can be relevant, to analyze if there is a particular instrument that can be more suitable to the anatomical orofacial features of the musician. Obtaining dental casts to study the occlusion of the wind instrumentalist or even as a preventive measure in the case of an eventual orofacial trauma can be an adequate routine procedure to implement in children that initiate a musical activity by the age of 7–8 years old.

Franz et al. had the objective to study the facial muscle activity which is crucial to musical performance in wind instrument playing. The mean electromyographic values were significantly higher in the students for the masseter, buccinator, and mylohyoid, while they were significantly higher in the postgraduates and professionals for the mentalis [20]. These preliminary data reflect a significantly higher overall facial muscle activity in the less-experienced group, potentially resulting in an overload, whereas the more expert players had more optimized muscle activity patterns [20].

The activity levels of orbicularis oris and digastric muscles were larger when playing in high tone than when playing in tuning tone in the brass instrument group [21]. These changes are also thought to be involved in the movement of angles of the mouth and back-downward force to the mandible [21].

Gotouda et al. did not find that playing an instrument for a long time induces fatigue of jaw-closing muscles [21]. From our perspective this can be analyzed with the assumption that wind instrument player has a very high tolerance to pain or to resistance regarding the muscle activity of certain muscles of the CCMC. As an example, on another study carried out by Clemente et al., it was possible to observe that the prevalence of asymmetric thermal patterns of these anatomical areas was not so high, which can till certain extent mean that these individuals’ area highly trained and experienced to the overuse of these anatomical sites [36]. Till certain extent this can be related to the EMG findings of Gotouda research, however there are many other investigations that relate the muscle hyperactivity present in wind instrumentalist performance as a predisposing factor of TMD. This topic of temporomandibular disorders and its relationship with playing a musical instrument, requires more studies in the future, since there are still researchers that confirm its correlation [14,16,18,37,38] while others state that there is poor evidence regarding this theme [15,17]. The reason for this to happen may be the implementation of different methodologies on the evaluation of TMD and musicians. Gotouda stated that the information obtained in wind instrumentalists’ examinations are useful as a stomatognathic function data including parameters such as playing career, conditions of dentition and occlusion, and presence of TMD symptoms, we agree with this observation and our research intends to clarify the issue related to the tooth position of wind instruments players, using a complementary method of diagnose very common in dentistry, namely in the area of orthodontics the cephalometric analysis.

Playing a single-reed instrument can exert horizontal and vertical forces on the maxillary and mandibular incisors that might result in maxillary incisor pro-clination, mandibular incisor retro-clination, and intrusion of maxillary and mandibular incisors, and therefore an increase in overjet and a reduction in overbite. From the selected studies, it appeared that single-reed players may have larger overjets compared to controls. Van der Weijden et al. did a systematic review on the topic regarding the influence of tooth position on wind instrumentalists’ performance and embouchure comfort [23]. From 54 papers, only two met the inclusion criteria, where a descriptive analysis showed that there are indications that tooth irregularities have a negative influence on embouchure comfort and performance of a wind instrument player [23]. The van der Weijden et al. study suggests that a Class I relationship without malocclusion seems appropriate for every type of wind instrument, while the more extreme the malocclusion, the greater the interference will be for the instrumentalists’ performance and embouchure [23]. On another study, van der Weijden et al. underwent a descriptive analysis indicated that adults playing a single-reed instrument may have a larger overjet than controls and that playing a brass instrument was associated with a reduction in overjet among children, which could be substantiated in a meta-analysis [24]. However, as it was possible to observe in the results of our study, this is not in accordance to most of the brass instrument players with the large cup-mouthpiece of the tuba players, presenting an overjet value of 4.18 mm, where the single reed instrumentalists presented overjet values of 4.66 and 4.06 mm, respectively. Studies from the systematic review and meta-analysis made by van der Weijden et al. mentioned single-reed players may have larger overjets as compared to controls [24]. This may happen due to the fact that playing a single-reed instrument can exert horizontal and vertical forces on the maxillary and mandibular incisors that might result in maxillary incisor pro-clination, mandibular incisor retro-clination, and intrusion of maxillary and mandibular incisors, and therefore an increase in overjet and a reduction in overbite [24].

Van der Weijden et al., suggest for future research, which is to inquire with the wind instrumentalists how they came about to choose their specific instrument of choice and whether their individual tooth position contributed to their choice [24]. From our perspective this is extremely relevant, however, in our opinion the area of health dental sciences, in particular dentists with the specialty and background of orthodontics or occlusion could be an active part on the analysis of the dentofacial profile of the wind instrument student that starts to choose his main instrument by the age of 6 or 7 years old.

This triad between wind instrument player, music teacher, and dentist, will certainly be more common in the future. In the sequence of the valid contribution that many researchers have done in the past and will continue to do in the future with the intention to analyze the embouchure mechanism, the skeletal morphology, the tooth position, the temporomandibular joint biomechanics, and the occlusion among other features of the CCMC of the musician, it will probably be possible and accurate to determine which wind instrument is more suitable for each particular musician.

Longitudinal studies were made by Brattstorn et al. being possible to find out that the cup shaped mouthpiece of the trumpet retro-clines and the single-reed mouthpiece of the clarinet pro-clines the maxillary incisors [28]. Rindisbacher et al. also had the intention to verify the influence on tooth position from playing a wind instrument, and the study was comprised of 62 musicians, 51 were music students in Berne, Switzerland, and 11 played in an orchestra or were music teachers, all played their instruments professionally [30]. The musicians were divided into two groups: the “brass instrument group” was comprised of 29 men and two women (12 trumpet, 10 trombone, and 9 French horn players), while the reed and flute instrument group was comprised of 25 men and 6 women (13 clarinet, 10 oboe or bassoon, and 8 flute players) [30]. The control group consisted of 75 men who did not play wind instruments [30]. It was possible to observe an overbite value smaller in the musician’s group and no difference in the overjet [30].

Shimada had already found out that it seemed worthwhile to study what kind of effect the wind instruments would exert in terms of morphology on the dento-oral region of the young people in the growing stage from a clinical orthodontic point of view, since there was an increase in the number of musical practitioners [31]. Interesting to notice was that many years ago, this author following the classification wind instruments made by Strayer (1939) selected a group of wind instrumentalists, however a subdivision of the Class A-cup shaped mouthpieces was made, with a Class A (S) being designated for those having small-cup shaped mouthpieces and Class A (L) for those having large cup-shape mouthpieces. In our perspective this was an important point of view regarding the classification for wind instruments made by Strayer and as a logical relation between the contact point of the wind instrumentalist mouthpiece and the oral facial structures Clemente et al., provide a new classification for wind instruments. Because if it is true that within the brass instruments, there are small-cup shaped and large cup-shaped mouthpieces it is also worthy to notice that there are visible differences existing with the angle of insertion of the singled reed and double reed of wind instrument players inside the oral cavity, which can, effectively or not, alter the tooth position of wind instrument players. The chance of observing with criteria all these determinants can be useful to gather more knowledge regarding these orofacial considerations and its relationship with musical performance. The findings from the cephalograms led Shimada to the following conclusions, that for the skeletal pattern, there was seen no obvious effect among the different groups of wind instruments, which is also in line with the recent publication of Clement et al. Regarding the denture pattern, Shimada observed mainly in the inclination angles of maxillary and mandibular anterior teeth depending upon the wind instruments used. Shimada stated that the main problem associated to the denture pattern is the inclination of maxillary and mandibular anterior teeth, and that any discussion of wind instrument players in this respect is directed to this problem. This was one of the main reasons that our investigation intended to address the tooth position in wind instrument players and comparing our results with Shimada, there can be some similarities. Shimada found out that for angle measurement, although there was observed no lingual inclination of Class A (S) maxillary and mandibular anterior teeth, there was found a slight tendency in Class A (L). In Class B, however, there was observed the labial inclination of maxillary anterior teeth on the one hand and mandibular anterior teeth were inclined lingually, on the other [31]. This finding may be explained by the fact that this type of wind instrument exerts much pressure on mandibular anterior teeth in the lingual direction [31]. In Class C and D alike, the labial inclinations of both maxillary and mandibular anterior teeth were confirmed. Measurements were also made on plaster models to find out whether any effect had been exerted on molar portion of the dental pattern. In terms of the present data, the sums of crown mesiodistal width of the male maxillary and mandibular were much larger than the mean of normal subjects [31]. Shimada explanation for this occurrence is related to the contraction of muscles in the mouth corner and cervical region, since these may be thought to exert a certain amount of influence on the players of various wind instruments [31].

To understand the existing difference of linear or angular measurements that make part of the dentofacial analysis of wind instrument players and the respective tooth position it is important to recognize specific patterns of certain malocclusions. Barbosa et al. evaluated the graniofacial growth of subjects with untreated Class II Division 2 malocclusion with a mixed longitudinal sample of 39 white Class II Division 2 subjects, analyzed at 5 distinct phases, namely at the age of 6–7, 9–10, 12–13, 15–16, and 18–19 years. From the Barbosa et al. study it was possible to conclude that subjects with Class II Division 2 malocclusion are more hypodivergent and have more upright maxillary incisors than do subjects with Class I occlusion [39]. Hypodivergence establishes itself early and increases progressively through early adulthood; maxillary incisor retro-clination occurs early [39]. Subjects with Class II Division 2 malocclusion demonstrated larger interincisal angles [39]. The interincisal angle decreased rapidly in both groups between 6 and 10 years of age, remained relatively stable through 13 years, and then increased slightly through adulthood. Analyzing in detail the evolution of the interincisal angle in 39 white Class II Division 2 and 35 subjects Class I control sample based on age and sex, it is interesting to notice some differences that however did not present statistically significance. At the time point age of 6–7 years, the Class II Div 2 group had an interincisal angle of almost 150° while the Class I had an interincisal angle value slightly above 140°. By the age of 9–10 years, the Class II Division 2 presented a value for the interincisal angle around 136° comparing to the approximate value of 131° for the Class I. At the time point of 12–13 years the Class II Division 2 is practically the same ±137°, while Class I seems to decrease the value of this dental variable to 130°. By the age of adulthood (18–19 years), the interincisal angle of the Class II Division 2 is of 140°, while the Class I presents a value around 133°. From our point of view this information is particularly interesting since the time points involved at Barbosa et al. investigation, correspond to important phase, 6–7 years old in which a young instrument player starts to choose his main and principal instrument. By the age of 12–13 years there is a high demanding increase in level in the study and performance of a wind or string instrument player. When comparing this data with the interincisal angles obtained in our sample of the 47 wind instrumentalists, which presented an average value of 127.50° and the group of string instrumentalist (*n =* 24) was of 122.34°, being this percentage of variability statistically significant (*p* < 0.05) for this measured parameter, it is worthy to question that if there was no equilibrium between the existing extra-oral forces and intra-oral forces of a wind player the interincisal angle would not be within the norm 130° ± 6.

In a previous study, the authors confirmed that within the group of wind instrument players, the majority presented their lower central incisor ortho-positioned or retro-inclined [40]. Therefore, it is our opinion that a wind instrumentalist on the orofacial region generally applies a higher pressure on the lower jaw, the only exception in this analysis was shown to be done by the clarinet player that induces a higher pressure on the upper jaw (central incisors) when stabilizing the mouthpiece in the oral cavity [32]. One of the possible explanations is related with the fact that the mandible being a mobile jaw/bone, moves upwards and downwards, executes protrusion and retrusion movements, while the mouthpiece is partly placed or adapted to this region even during these movements inherent to the TMJ biomechanics. This happens during the musical performance of a wind instrumentalists where these movements of the lower jaw occur to allow different kinds of pitches, while obtaining low, medium, and high notes. The contact point of the mouthpiece is on the upper jaw, independently to the fact that this can occur on the lips or on the teeth, happens with the intention of stabilizing the wind instrument on the anterior zone that can be considered a support zone.

Our study presents some limitations regarding the sample, which can represent a low statistical power, however it is worthy to notice that this research involves 48 wind instrument players and 24 string instrument players, which is in one of the largest sample analyzed concerning a cephalometric analysis, also in accordance to previous studies [28,29,30,31]. Nevertheless, a descriptive analysis will be done along this discussion of the small sample of musicians analyzed within the wind instrumentalists, where it is normal to have more for example more clarinet (*n* = 10), saxophone (*n* = 8), or trumpet players (*n* = 10), in comparison with double reed instrument players (*n* = 4), since this reflects the number of students that are attending musical classes in a superior school of music.

A research that involved 12 trumpeters and 12 clarinetists, aged 19–55 years, were compared with a control group of dental students [29]. The pressures recorded during playing of the instrument were considerably greater than those found during both chewing and speech. Despite this, no effect on the dentition was found [29]. Fuhrimann et al., explanation for the fact that all three groups were rather similar with respect to facial and bite morphology, which may be related with the fact that professional musicians, like the subjects studied, often play several wind instruments, the influence of which on the dentition may be in different directions [29]. It must also be taken into consideration that the groups studied were small and hence did not reveal possible small morphological differences between them [29].

Regarding the dentoskeletal morphology in adults with different Class II Division 1 or Class II Division 2 malocclusion with increased overbite, Deniz Uzuner et al. revealed in their study that there are significant differences in the maxillary and mandibular dentoalveolar morphology among the increased overbite groups (overbite 4.5 mm) and the control group [41]. Regarding incisor inclinations, significant differences were found between groups [41]. The Class I increased overbite group had rather normal inclination of maxillary incisors whereas the mandibular incisors were found to be retrusive and retro-clined [41]. In the Class II/1 group, the lower incisors were pro-clined and the inter-incisal angle was reduced, which may be due to compensation for the increased overjet [41]. In the Class II/2 group, however, the maxillary and mandibular incisors were retrusive, and the interincisal angle was significantly increased [41]. The differences between groups were related primarily to inclinations and vertical positions of the incisors, rather than molar positions [41]. According to other authors the overbite changes along the craniofacial growth, since it decreases with the vertical growth of the mandibular ramus, the eruption of the second molars and increases with the mesialization of the molars [42,43,44,45]. However, taking in consideration that one of the main dental features in the cephalometric analysis of the increased overbite can be the occlusal contact point of the incisors, it is interesting to observe based on our results that the trumpet and the French horn are the wind instrumentalists that show a smaller value for the lower incisor parameter, −0.33 and −0.36 mm, respectively, and present overbite values with more than 3 mm. While other authors report that the increased overbite may be associated to the supra-eruption of mandibular incisors as a determinant factor [46,47,48,49], it was notorious that within our sample of eight saxophone players these had an overbite value of 1.54 mm, below the norm of 2 mm. This can be related to the position adopted by the mouthpiece inside the oral cavity, since this occurs in a more parallel manner according to the Frankfurt plan [50], while the forces can be considered intrusive—in the apical direction of the lower central incisors—and are actually one of the higher values analyzed within the single reed instrumentalists [51]. The results of Uzenur et al., with the interpretation that the influence of lip pressure may be another factor in increased retro-clination, can eventually be linked to the previous finding of Clemente et al. that, from the evaluation of craniofacial morphology of wind and string instrument players, found that the only statistical difference was lower incisor inclination (*p* = 0.011), concluding that playing a wind instrument showed to have little orthopedic influence at the craniofacial morphology, on contrary it may influence the lower incisor inclination with its osseous base [40].

Another interesting factor is the eventual impact that the tongue position can have on the equilibrium of the intra-oral forces applied against the extra-oral forces of the mouthpiece towards the oral facial region in particular the brass instrumentalists. Adesina et al. refer that tongue size, posture, and pressure are considered to have significant influences on the positioning of dentoalveolar structures [52]. This study found tongue measurements (thickness, length) to be generally higher in bimaxillary pro-clination [52]. Analyzing wind instrument players embouchure mechanism, it is possible to understand that there is an impact of the instrument executing a direct or indirect pressure on the lower and upper lip (tuba, trombone, French horn, trumpet, bassoon and oboe), or on the lower lip and upper teeth (clarinet and saxophone) which may explain the force vector that originates a reduced interincisal angle. This is also in agreement of our findings related with the lower incisor protrusion, as the transverse flute appears with the highest value within the wind instrument group, 4.23 mm.

Kirschneck et al. carried out a retrospective cephalometric study on the association of dentoskeletal morphology with incisor inclination in angle class II patients and concluded that the lower incisor pro-clination significantly decreased from groups Class II/1 and II toward group Class II/2 an the interincisal angle was found to increase continuously from group Class II/1 to group Class II/2. Interesting to notice that these facts presented by Kirschneck et al. regarding the magnitude of the interincisal angle which is associated with the extent of vertical overbite, particularly in Class II division 2 patients, in some kind does not match with the parameters that were found within our wind instrumentalist group, since the French horn group had the higher value of overbite, a mean value of 3.20 mm (*n* = 3 French horn players) and a interincisal angle of 125.90°, not corresponding at all with the highest value for the interincisal angle which belonged to the bassoon players, with a mean value of 133.94°. However, the existence of high interincisal values were found on a previous study by Clemente et al., with a cephalometric analysis of a bassoon wind instrument player showing values of 145.10°, with both maxillary and mandibular incisors in a retro-clined position, upper incisor had 20.2° (value expected: 28 ± 4°) and the lower incisor had 14.7° (value expected: 22 ± 4°) [53]. Understanding the embouchure mechanism is fundamental in dental health sciences, since when a dentist is analyzing the embouchure of a bassoon player it is important to notice that both the upper and lower lips are retruded over the teeth carrying out undesired forces. On the other hand, for example, in the case of the French horn player that didn’t present an interincisal angle so high, appeared with higher values regarding the overbite, which can be related to higher pressures applied on the upper and lower lip, that can reach up to maximum values of 220 and 305 gf, correspondently [32].

An exaggerated curve of Spee can represent one of the highest contributing components among all the dental and skeletal ones for the main role in the development of deep overbite [54]. The augmented curve of Spee showed a prevalence of 78% and a decreased gonial angle as a skeletal reference were the greatest contributing components for the deep overbite occlusion, whereas the retro-clination of the mandibular incisors (21.8%) was one of the least contributing factors [54]. The correlation of El-Dawlatly et al. study [54] with our results can be expressed within the wind instrument group, where only the saxophone players (*n* = 8) did not present a value above the 2 mm (norm) for the overbite parameter, presenting an average value of 1.54 mm. This can be interrelated with the intrusive forces that are applied by the saxophone mouthpiece on the lower lip that is retruded over the mandibular incisal edge, preventing any extruding movement of the mandibular anterior segment which would necessarily influence the curve of Spee.

Zupanic et al. refer that in Class II division1 malocclusion subjects, there is another parameter which can be a significant predictor, the overjet. Regarding this aspect, pointed out by Zupanic et al., it is interesting to notice that dental relationships in wind instrument players will oblige the adoption of different skeletal relationships in the sagittal plane during musical performance, more specifically at the embouchure mechanism [55]. It can be observed on the study of Clemente et al., that the cross-sectional angle between the Frankfurt horizontal plane and a line traced at the superior part of the brass mouthpiece, more specifically in the contact point of the embouchure where the mouthpiece adapts to the orofacial region, there is an existing difference between the trumpet and tuba players, with an average of 71.6° and 81.5° [50]. The interpretation for this occurrence can possibly be associated to the anterior tooth position of these wind instrument players. The results of our research confirm these findings since the overjet values that was found on the brass instruments of large cup-mouthpiece were within the higher values obtained. The overjet value for the trombone group was of 5.06 mm, while the tuba group presented 4.18 mm. There was an exception in the small cup-mouthpiece, the French horn group that appeared with an overjet value of 6.18 mm. This last case can be related as it was mentioned before with the high force values that are executed and eventually also the size and weight of the instrument, which will oblige the French horn player to adjust their embouchure mechanism in different modes differing from player to player what would be a regular embouchure. These are normal variations that can occur, because in the authors opinion the factors that can influence the above mentioned overjet of the large cup-mouthpiece when comparing for example with the small cup-mouthpiece like trumpet group (*n* = 10) is a positive correlation between the contact point of the mouthpiece, since in the case of the tuba and trombone this occurs in the maxilla at a subnasal point where the nasal septum merges with the upper cutaneous lip in the midsagittal plane, which can induce a pressure in the root of the central incisors originating an axis of rotation of the teeth that will promote the pro-inclination of the teeth with the overjet values that were obtained in our results. While in the case of the trumpet the overjet value is lower, 3.88 mm, since the contact point of the small cup-mouthpiece is centered more at the lip corresponding to the crown of the upper and lower incisors. Curiously when analyzing the applied forces on the perioral structures of 6 trumpet players, Clemente et al. found out that the mean forces for the maximum values applied where higher on the lower lip with 172 gf, compared to the 130.3 gf on the upper lip. The maximum values for the lower lip in the trumpet player reached a value of almost 300 gf compared to the 201 gf of the upper lip [32]. This data can be associated to our results, and possibly reflect the variations that are obtained for the lower incisor protrusion for the trumpets with −0.33 mm, comparing for example with the tuba that had a 4.18 mm. Once more the tuba average value for the lower incisor protrusion can be correlated to the contact point where the large cup-mouthpiece rests against the soft tissue below the lower lip corresponding to a supramental region, a deepest point on the outer contour of the mandible. Once more the contact point pressure applied in this area can explain the high value of the lower incisor protrusion for the tuba.

More than dento-alveolar compensations the musicians embouchure mechanism show that there are extra-oral forces applied by the brass mouthpieces, while the intra-oral forces are applied by the tongue and the mouthpiece of double and single reed mouthpiece instruments. The different anatomical site of the contact point will make a difference in the final tooth position of the central incisors. As an example, we can observe from our results the overjet of the clarinet group (*n* = 10) with a value of 4.66 mm, while the saxophone group (*n* = 8) presented a overjet value of 4.06 mm, when the norm is 2 mm. The reason for this occurrence can be probably associated to the fact that the insertion of the clarinet inside the oral cavity occurs in a more vertical manner, when compared for example with the saxophone. This implies a higher pressure on the palatal surface of upper central incisors of the clarinet group. Instead of this the saxophone induces a higher pressure on the lower lip that is retruded over the mandible incisors, where the average force (median values) of five clarinet players is 58.8 gf, in contrast to the saxophonists that can reach 94 gf on the lower lip. This observation of the tooth position adopted in the clarinet group with an overjet higher than the saxophone group, is in agreement with the force measurement parameters for the applied forces on the upper incisors in a recent study that demonstrated the mean maximum value of 106 and 82.7 gf, respectively [51].

Ceylan et al., highlights that there are dentoalveolar compensatory changes in the position and axial inclination of the maxillary and mandibular incisors related to the variations in the overjet pattern [56]. The values of the interincisal angles in the study were considered as follows, normal—129.6°, edge to edge—132.1°, negative—Class II Div 2—138.8°, positive—Class I Div 1— 127.6° [56]. It was possible to find that the axial inclination of the mandibular incisors in subjects with normal and positive (increased) overjet were similar, whereas lower incisor inclinations in the negative (decreased) overjet subjects were significantly different from those of the normal and positive overjet cases [56]. The lower incisors in the negative overjet cases were more upright than in the other overjet groups. The upper and lower incisors inclined more labially in the positive than in the negative overjet cases [56].

The presence of parafunctional habits such as digit-sucking has been reports in the literature as a possible etiologic factor of a malocclusion. Sushmitha Singh et al. evaluated dentoskeletal characteristics in a group with digit-sucking that was submitted to a cephalometric study [57]. The angle between mandibular incisors and mandibular plane (MNI-MNP) reduced statistically in the study group (99.17°) when compared to the control group (105.33°), depicting accentuated retro-clination of mandibular incisors [57]. Maxillary incisors were significantly pro-clined relative to the cranial base (MXI-SNL) in the study group (109.47°) when compared to the control group (95.40°); this might be due to the lever effect of the digit creating an anteriorly directed force on the maxillary alveolar process and incisors [57].

Digit-sucking is proven to cause adverse effects on occlusion and dentition, with most common effects being: anterior open bite, maxillary bone and dentition narrowing, protraction of anterior teeth and premaxilla, and crowding of mandibular incisors. Teeth deformation and the alveolar processes exhibit a configuration that is a negative impression of the thumb [58]. Muscular activity is also abnormal, with hyperactivity of the buccinators muscles that contribute to the narrowing of the palate [59,60]. A previous study reported that in patients with no or small mandibular movement both maxillary and mandibular incisors have labial inclination while in patients with greater mandibular movement, the dental axes of mandibular incisors are upright [19].

Deleterious oral habits are automatic and often unconscious actions, presenting an altered pattern of muscle contraction. These habits are acquired by practicing a nonfunctional action, are usually seen in infancy, and most of the times start and stop spontaneously, without any consequence [61].

Physiologic sucking behaviors, like breastfeeding, are considered normal habits in newborns once they contribute for the normal development of the craniofacial structure: it stimulates orofacial muscles and contributes to normal growth. However, as the child grows, the habit should disappear [61]. When deleterious habits like pacifier or thumb sucking persists in time, this can result in dental and skeletal alterations due to the imbalance between external and internal muscles and the pressure of the thumb, lip, or tongue, but can be successfully treated by an early interceptive intervention. If the habit continues while the permanent dentition is becoming established, it can cause the development of a malocclusion [62]. Numerous studies have linked non-nutritive sucking habits to malocclusion [63] where the development of an anterior open bite is common to occur [62,64]. In order to play a wind instrument the musician has to make a propulsion movement of the mandible repeatedly, similar to the mandibular movement of the patients who have a thumb sucking habit with propulsion. We can justify the common morphological change of the lower incisors up righting with the similar back and forward movement. The main difference found between the two is that the finger sucking patients usually present a protraction of the upper incisors and wind instrument players do not. This can be explained by the muscle hyperactivity showed in musicians, with hypertonic lips developed by the hyperactivity of the orbicular oris during musical performance [40].

Deleterious oral habits are automatic and often unconscious actions, presenting an altered pattern of muscle contraction. These habits are acquired by practicing a nonfunctional action, are usually seen in infancy, and most of the times start and stop spontaneously, without any consequence. Physiologic sucking behaviors, like breastfeeding, are considered normal habits in newborns once they contribute for the normal development of the craniofacial structure: it stimulates orofacial muscles and contributes to normal growth. However, as the child grows, the habit should disappear [61]. When deleterious habits like pacifier or thumb sucking persist in time, this can result in dental and skeletal alterations due to the imbalance between external and internal muscles and the pressure of the thumb, lip, or tongue but can be successfully treated by an early interceptive intervention. If the habit continues while the permanent dentition is becoming established, it can cause the development of a malocclusion [62].

The presence of parafunctional habits such as digit-sucking has been reported in the literature as a possible etiologic factor of a malocclusion. Digit-sucking is proven to cause adverse effects on occlusion and dentition, with most common effects being: anterior open bite, maxillary bone and dentition narrowing, protraction of anterior teeth and premaxilla, and crowding of mandibular incisors [61].

Muscular activity is also abnormal, with hyperactivity of the buccinators muscles that contribute to the narrowing of the palate [59,60]. In order to play a wind instrument the musician has to make a propulsion movement of the mandible repeatedly, similar to the mandibular movement of the patients who have thumb sucking habit with propulsion. We can justify the common morphological change of the lower incisors up righting with the similar back and forward movement. The main difference found between the two is that the finger sucking patients usually present a protraction of the upper incisors and wind instrument players do not. This can be explained by the muscle hyperactivity shown in musicians, with hypertonic lips developed by the hyperactivity of the orbicularis oris muscle during musical performance and by the pressure exerted by the mouthpiece, especially on brass musicians [40]. This contradicts the protrusion movement of the upper incisors. Brattstorm et al. reported that musicians had a decreased anterior facial height and wider dental arches, which was interpreted with an increased muscle activity and intra-oral pressure resulting from wind instrument playing [28]. In a study by Engelman et al., it has been proven that the instrument pressure was less than the thumb-sucking pressure and higher than swallowing and whistling [65].

On the other hand, sleep-related breathing disorders in children with malocclusion have also be considered as another important casual factor for the development of a malocclusion, where patients with sleep-related breathing disorders (SRBD) had a smaller maxillary width (*p* < 0.003), and according to the cephalometric study, less overbite (*p* < 0.003) [66]. Furthermore, the prevalence of sleep-related breath disorders was higher among patients with a history of adenotonsillectomy (*p* < 0.02) [66].

Oral breathing and atypical deglutition [67] are another type of deleterious oral habits worth of mentioning and comparison. Oral breathers usually present a nasal obstruction which leads to mouth breathing, resulting in a change of the tongue’s position and half opened lips [62]. Ricketts has attributed a reduction in SNB angles in cases of nasal obstructions to a more forward and downward tongue posture to facilitate oral breathing [68]. A previous study that measured levels of the bioelectric potential activity in the masseter and anterior temporal muscles in oral and nose breathers found that activity was lower among mouth breathers. Some of the alterations found in mouth breathers are hypotonic and hypofunctional jaw elevator muscles and ineffective mastication because of jaw elevator muscle laxity or even by poor coordination between breathing, mastication, and swallowing [60].

In establishing a comparison between these parafunctional habits and playing a wind instrument, there is a need to understand why the first can cause serious dental alterations and the second does not necessarily lead to orthopedic alterations at the craniofacial morphology, but still present a high prevalence of dentofacial parameters that are relevant to mention and contextualize in the area of performing arts medicine. The authors of this study believe that the possible explanation associated to the fact that most of the wind instrumentalists present an increased overjet and an increased overbite, while the patients that have a deleterious habit of digit-sucking, of using pacifiers or present atypical deglutition have an increased overjet and diminished overbite is related to the fact that in these last parafunctional activities the pressures and forces that are induced in the perioral region are originated inside the mouth and the vector of forces is objectively directed towards the outside of the oral cavity. On the other hand, the induced pressures and forces that are applied on the perioral regions of wind instrumentalists diverse in the origin and direction of the force vectors. In the case of wind instrument players, the forces are applied by the mouthpiece inducing in most part of the cases a force −0.3 against the perioral tissues, with a direction of forces towards the inside of the mouth.

What really happens after applying this force in the direction of the oral cavity is different, since the vector of forces and the results of these forces will diverse due to the existence of different contact points of the mouthpiece. As a result, in the case of wind instrument players the external forces directed towards the perioral region can pro-incline the anterior teeth, since the large cup-shaped mouthpiece for example have a fulcrum on the area corresponding to the root of the upper central incisors, which will necessarily change the point of rotation center, promoting the increased overjet. In the case of single reed instruments, with different insertion angles of the mouthpiece inside the mouth, there is always a pressure applied towards the palatal and incisal edge of the upper central incisors which will necessarily provide the existing forces for the natural movement of pro-inclination of these teeth, with the appearance of the overjet parameter.

The high prevalence of the overbite parameter in the wind instrumentalists occurs in the sequence of the muscular hyperactivity that is promoted at the orbicularis oris and in the line of thought of the authors, this occurs on all wind instrument players even if the reasons are distinct. While the brass instrumentalists have an hyperactivity of the lips associated to the lip vibration, the single reed instrument player have also hypertonic lips due to the necessary sealing of the orbicularis oris muscle, namely the superior lip, around the mouthpiece for the column of air not to “leak” or escape outside the mouth (since the air flow has to be directed to mouthpiece in order to promote the reed vibration and the inherent sound production), while in the case of double reed instrument players also present a high activity of the lips when these retrude over the incisor edge over the upper and lower incisors, providing in this particular way the necessary forces for the occurrence of overjet.

In summary, while the deleterious habits of digit-sucking, using pacifiers or present atypical deglutition have forces from inside the oral cavity to the outside promoting the overjet, playing a wind instrument induces forces with a direction to the inside of the mouth, which can promote the occurrence of overjet and overbite, since there is a projection of the upper incisors at an anti-rotational direction (action of the mouthpiece) and at the same time with the lip action at a downwards direction.

The results of our study showed that the interincisal angle is statistically significant between the wind instrument group and the string instrument. The reason for this fact, can eventually be associated to the act of playing a musical instrument, which implies a demanding physical activity with the activation of distinct group of muscles from the CCMC. It is known that viola and violin players usually clench their teeth while supporting the string instrument with the chin against the shoulder and chest. This parafunctional habit of teeth clenching can eventually be responsible for an intensive contact of the lower incisal edge towards the palatal surface of the upper incisors, promoting the pro-inclination of the upper teeth, and consequently reducing the interincisal angle average to 122.34° comparing to the 127.46° of the wind instrumentalists group average. Even the 6 cello players presented an average value of 120.95° for the interincisal angle, being this value lower than most of the presented average of the trombone, tuba, trumpet, French horn, transverse flute clarinet, saxophone, and bassoon different wind instrument group. The reason for this can be linked to an analysis done, also with another string instrument, the piano, where the authors questioned “Do piano players play with their teeth?” In fact piano players did not play with their teeth, nevertheless as it is possible to observe, the bioelectric potential of the masseter muscles in a pianist presented a value of 49.6 µV while playing, for example, the C minor Rachmaninoff concerto, compared to the 15–18 µV of the same muscular group during the mastication process 61. Comparing a primary function of the masticatory muscles such as eating with a parafunctional activity like piano performance, it is possible to realize that string instrument players do induce forces within the CCMC, which can be in the origin of the above mentioned dentofacial parameters as documented in the results of our cephalometric analysis.

There are relatively few variations on the analyzed parameters overjet, overbite, lower facial angle, facial convexity between the wind and string instrument group, with the exception to the lower incisor protrusion that is higher in the string group with average values of 3.48 and 2.40 mm, comparatively to the wind group. This can be eventually associated to the fact that wind instrument players present an hyperfunction of the lower lip, since the lower lip during the embouchure is retruded over the lower incisors, when playing the clarinet, the saxophone, and the bassoon. In the cases of the brass instruments there is a major activity of the lower lip during the vibration movements that occur in synchronization with the upper lip. Therefore, the increased activity of the lower lip in wind instrumentalists was probably one of the factors that promoted a reduced lower incisor inclination with its osseous base, being more orthopositioned [40]. In the same line, the results of this investigation showed a reduced average value of 2.40 mm for the lower incisor protrusion when compared to the string instrument players since these musicians do not necessarily produce such activity on the lower lip during musical performance.

Independently to the origin or etiological factor that can act as a predisposing element for the development of a malocclusion, the work that is addressed along this manuscript represents an effort to analyze the tooth position in wind instrumentalist and string instrumentalist with the evidence and scientific support that is available till the moment. This dentofacial cephalometric analysis pretends to complement previous efforts made by other researchers in order to contribute to area of performing arts medicine, namely in the area of dental sciences. The triad between wind instrument player, music teacher, and dentist will certainly be more common in the future. In the sequence of the valid contribution that many researchers have done in the past and will continue to do in the future with the intention to analyze the embouchure mechanism, the skeletal morphology, the tooth position, the orbicularis oris muscle function, the temporomandibular joint biomechanics, and the occlusion among other features of the CCMC of the musician, it will probably be possible and accurate to determine which wind instrument is more suitable for each particular musician. It is important that dentistry literacy and related issues of performing arts can be a matter for future discussion, since it is an area where, objectively, dentists, maxillo-facial surgeons, musicians, musical pedagogues, and even the parents of young musicians that start a musical activity can bear in mind all these considerations related to the orofacial region, the embouchure mechanism, and the tooth position.

## 5. Conclusions

This study’s findings demonstrate that when evaluating the two samples, wind and string instruments, there are different dentofacial configurations, however the only statistically significant differences that were found are related to the interincisal angle (*p* < 0.05).

The knowledge of the overjet and overbite value permits a substantial analysis on the tooth position of wind instrument players, where both of these parameters are increased and greater than the norm value. The cephalometry was an added value on the interpretation of possible factors that lead to the position of the central incisors of wind instruments.

The applied forces during the embouchure mechanism will certainly influence the inclination of the anterior teeth, that to some extent present an equilibrium on the vector of forces. These are directed in most of the cases to the inside of the oral cavity by the mouthpiece and directed downwards due to the orbicularis oris—lips force. If such equilibrium did not take place there could be a higher value for the overjet parameter and a low value for the overbite.

The major point is that normally, when there is an overjet, the cause and forces are applied from the inside of the mouth, e.g., digital sucking, pacifiers, and atypical deglutition, where in this study we analyzed and confirmed the occurrence of this parameter with forces being applied from the outside of the mouth—mouthpiece.

Still, there are relatively few variations on the analyzed parameters overjet, overbite, lower facial angle, and facial convexity between the wind and string instrument group, with the exception to the lower incisor protrusion.

The data available on this study provides information regarding the tooth position that can eventually be taken in consideration by dentists, orthodontists, maxillo-facial surgeons, musicians, musical teachers, and parents when initiating musical performance and choosing a wind instrument.

## Figures and Tables

**Figure 1 ijerph-18-04306-f001:**
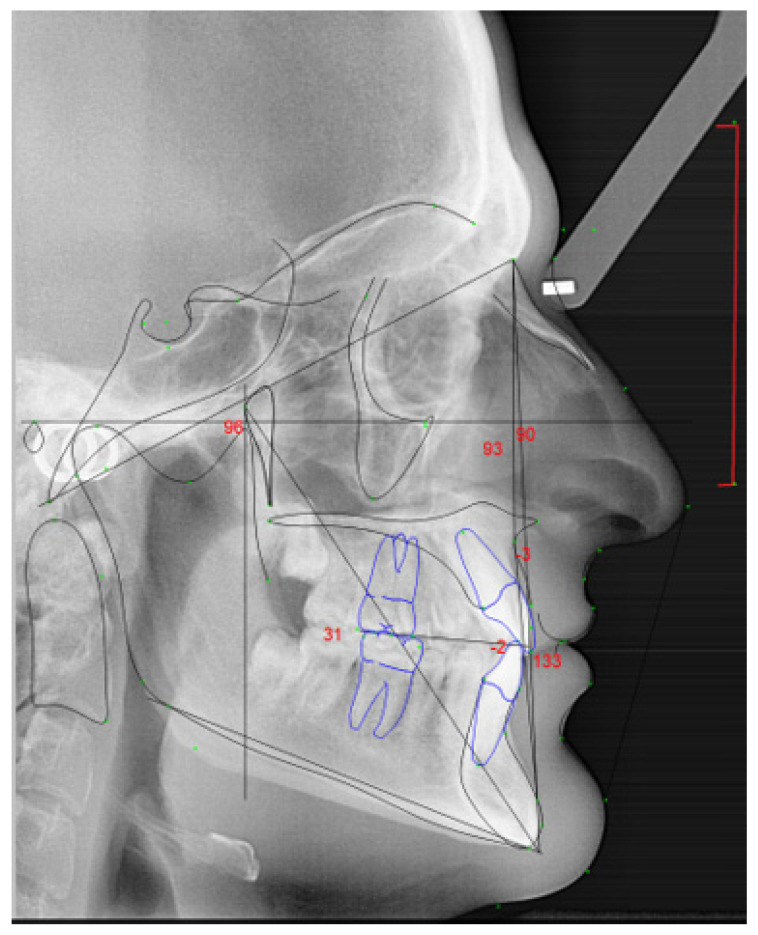
Lateral tele-radiography of the head for dentofacial cephalometric analysis.

**Figure 2 ijerph-18-04306-f002:**
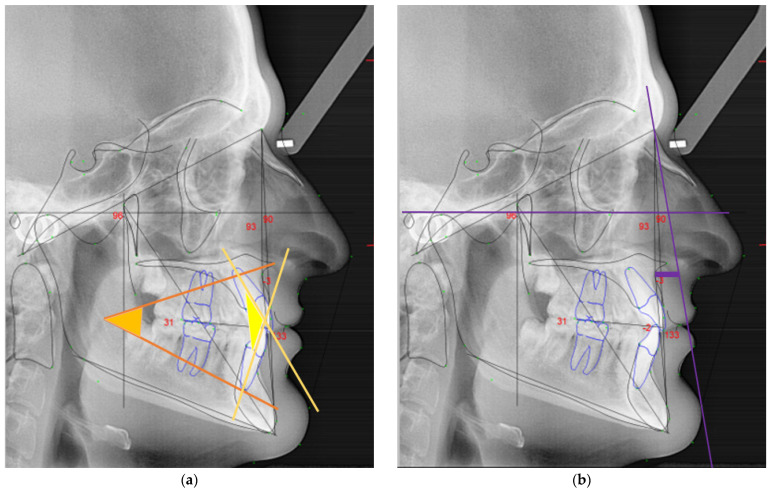
Rickett’s cephalometric analysis with specific parameters analyzed: (**a**) lower facial angle (orange) and interincisal angle (yellow); (**b**) facial convexity (purple).

**Figure 3 ijerph-18-04306-f003:**
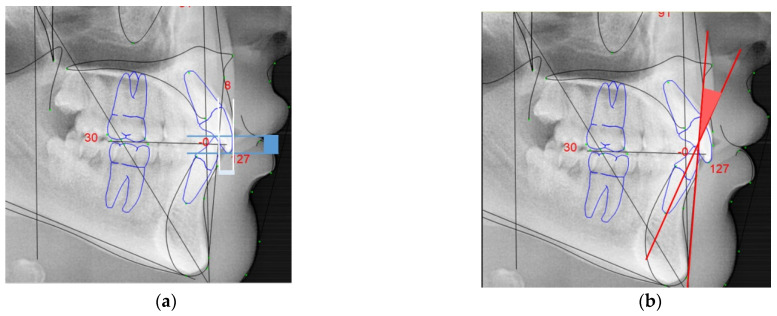
Linear and angular references for cephalometric analysis studied: (**a**) overbite (blue) and overjet (white); (**b**) lower incisor protrusion (red).

**Table 1 ijerph-18-04306-t001:** Descriptive statistics for the variable interincisal angle.

	Interincisal Angle	Norm	Average	Std. Dev.	Maximum	Minimum	Average Per Group
Wind	Clarinet (*n* = 10)	130° ± 6°	126.96	8.764	141.30	114.10	127.46
	Bassoon (*n* = 4)		133.94	8.188	145.10	126.20	
	Transverse flute (*n* = 5)		125.85	11.468	139.10	110.10	
	Saxophone (*n* = 8)		128.73	5.201	136.10	121.70	
	Trombone (*n* = 4)		127.72	15.026	146.60	107.10	
	French horn (*n* = 3)		124.63	7.663	132.80	117.60	
	Trumpet (*n* = 10)		129.83	10.995	149.90	113.00	
	Tuba (*n* = 4)		123.05	12.531	105.60	132.60	
String	Viola (*n* = 4)		113.92	10.008	105.80	130.30	122.34
	Violin (*n* = 14)		125.20	8.861	104.10	139.20	
	Cello (*n* = 6)		120.95	3.654	115.80	123.60	

**Table 2 ijerph-18-04306-t002:** Descriptive statistics for the variable overjet.

	Overjet	Norm	Average	Std. Dev.	Maximum	Minimum	Average Per Group
Wind	Clarinet (*n* = 10)	2 mm	4.66	1.056	6.00	3.50	4.05
	Bassoon (*n* = 4)		3.10	1.636	5.80	1.50	
	Transverse flute (*n* = 5)		3.55	1.905	6.10	1.70	
	Saxophone (*n* = 8)		4.06	1.361	5.90	2.50	
	Trombone (*n* = 4)		5.06	2.340	2.40	7.70	
	French horn (*n* = 3)		5.20	1.500	6.70	3.70	
	Trumpet (*n* = 10)		3.88	1.347	5.10	1.70	
	Tuba (*n* = 4)		4.18	0.862	4.70	2.90	
String	Viola (*n* = 4)		3.88	1.699	6.60	2.60	4.55
	Violin (*n* = 14)		4.72	1.394	6.90	2.50	
	Cello (*n* = 6)		4.98	1.090	5.90	3.60	

**Table 3 ijerph-18-04306-t003:** Descriptive statistics for the variable overbite.

	Overbite	Norm	Average	Std. Dev.	Maximum	Minimum	Average Per Group
Wind	Clarinet (*n* = 10)	2 mm	3.73	2.484	8.50	1.00	3.05
	Bassoon (*n* = 4)		2.60	1.789	0.80	5.60	
	Transverse flute (*n* = 5)		4.72	1.688	6.60	3.20	
	Saxophone (*n* = 8)		1.54	2.720	5.60	−2.70	
	Trombone (*n* = 4)		2.88	2.310	4.90	−0.80	
	French horn (*n* = 3)		3.20	0.436	3.50	2.70	
	Trumpet (*n* = 10)		3.58	3.161	−1.20	7.10	
	Tuba (*n* = 4)		4.33	1.797	2.00	6.10	
String	Viola (*n* = 4)		2.22	2.038	0.20	5.60	3.24
	Violin (*n* = 14)		3.37	2.633	8.50	−0.60	
	Cello (*n* = 6)		4.25	1.535	6.20	2.60	

**Table 4 ijerph-18-04306-t004:** Descriptive statistics for the variable lower facial angle.

	Lower Facial Angle	Norm	Average	Std. Dev.	Maximum	Minimum	Average Per Group
Wind	Clarinet (*n* = 10)	47° ± 4°	43.30	4.031	49.30	37.00	43.63
	Bassoon (*n* = 4)		45.74	2.038	48.20	43.60	
	Transverse flute (*n* = 5)		44.95	3.885	50.20	41.50	
	Saxophone (*n* = 8)		43.94	4.780	53.70	38.50	
	Trombone (*n* = 4)		43.08	3.703	48.80	38.80	
	French horn (*n* = 3)		43.93	4.119	46.70	39.20	
	Trumpet (*n* = 10)		42.64	5.020	50.80	33.00	
	Tuba (*n* = 4)		41.43	8.538	48.50	29.80	
String	Viola (*n* = 4)		43.42	3.185	47.70	39.10	42.10
	Violin (*n* = 14)		42.08	4.187	52.10	37.10	
	Cello (*n* = 6)		41.23	3.366	39.20	45.90	

**Table 5 ijerph-18-04306-t005:** Descriptive statistics for the variable facial convexity.

	Facial Convexity	Norm	Average	Std. Dev.	Maximum	Minimum	Average Per Group
Wind	Clarinet (*n* = 10)	2.0 mm	2.93	5.010	10.80	−6.80	3.08
	Bassoon (*n* = 4)		4.56	0.981	5.60	3.50	
	Transverse flute (*n* = 5)		4.05	4.090	9.10	−0.90	
	Saxophone (*n* = 8)		0.66	5.960	10.80	−6.50	
	Trombone (*n* = 4)		1.58	2.640	4.50	−0.80	
	French horn (*n* = 3)		0.13	1.401	1.70	−1.00	
	Trumpet (*n* = 10)		3.53	4.319	8.00	−5.60	
	Tuba (*n* = 4)		5.00	4.760	9.20	−1.80	
String	Viola (*n* = 4)		4.84	3.334	9.60	1.40	3.54
	Violin (*n* = 14)		3.20	3.579	10.70	−2.70	
	Cello (*n* = 6)		4.65	4.359	11.00	1.70	

**Table 6 ijerph-18-04306-t006:** Descriptive statistics for the variable lower incisor protrusion.

	Lower Incisor Protrusion	Norm	Average	Std. Dev.	Maximum	Minimum	Average Per Group
Wind	Clarinet (*n* = 10)	1.0 mm	3.92	3.335	8.80	−2.80	2.40
	Bassoon (*n* = 4)		1.94	7.278	14.70	−3.50	
	Transverse flute (*n* = 5)		4.23	3.494	8.80	0.30	
	Saxophone (*n* = 8)		2.36	1.339	4.80	0.60	
	Trombone (*n* = 4)		2.92	2.327	5.10	−0.40	
	French horn (*n* = 3)		−0.367	5.572	2.90	−6.80	
	Trumpet (*n* = 10)		−0.33	3.093	5.80	−3.60	
	Tuba (*n* = 4)		4.18	4.252	10.30	0.70	
String	Viola (*n* = 4)		5.60	2.545	7.50	1.20	3.48
	Violin (*n* = 14)		2.74	3.006	7.30	−2.70	
	Cello (*n* = 6)		4.48	9.580	18.60	−2.70	

**Table 7 ijerph-18-04306-t007:** *p* results for the independent *t*-tests comparing string and wind groups.

Parameters Tested	Sig. (2-Tailed) = *p*
Overjet	0.199
Overbite	0.765
Lower facial height	0.154
Facial Convexity	0.664
Lower incisor inclination	0.309
Interincisal angle	0.039

**Table 8 ijerph-18-04306-t008:** *p* results for the independent t-tests comparing metal and woodwind groups.

Parameters Tested	Sig. (2-Tailed) = *p*
Overjet	0.328
Overbite	0.070
Lower facial height	0.361
Facial Convexity	0.959
Lower incisor inclination	0.252
Interincisal angle	0.774

## Data Availability

Data are available upon request to the correspondent author.

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
