# Peer review of "Tooth Position in Wind Instrument Players: Dentofacial Cephalometric Analysis"

_ijerph, 2021, doi:10.3390/ijerph18084306_

Round 1
Reviewer 1 Report
Please explain why average numbers in the tables are only restricted to certain values? Please do all with standard deviations.
Please explain different statements for clarity like this one for example (it is relevant to contextualize the area of dental sciences 184 regarding a sub-speciality that can be considered, performing arts medicine)
Author Response
Attched

Reviewer 2 Report
Manuscript ID: ijerph-1154705
Tooth position in wind instrument players: cephalometric analysis
While the overall idea for this engineering and research effort is good, the presentation of it in the paper minor substantial improvement.
Abstract
- Is there a positive correlation between the way of performing and oral diseases?
2.“Specific dentofacial characteristics in wind instrumentalists should be taken in consideration when analysing physiological and anatomical issues regarding, the musician’s embouchure, posture and biomechanics during musical performance.”
What is the purpose of this research? Improve the way of performance?
- As a result, there are no major discoveries or suggestions for improvement? The bright spots should be highlighted.
Introduction
- “Playing related musculoskeletal disorders (PRMD) have been described (3, 12, 13) 49 and also the association between playing a musical instrument and the prevalence of tem-50 poromandibular disorders (14-18).”
This should be stated in the Abstract.
- I think that wind instrument players and string players cannot be compared. Because the two ways of playing are very different.
Materials and Methods
- The experiment is only limited to more than 10 years of performance, but there is no difference in age. I think this must be improved. Because if it is someone who has performed for 10 or 20 years, it is twice as bad.
- How do I calculate the number of participants in the experiment? “This study involved 48 wind instrumentalists (67%) and 24 string instrumentalists (33%) from the Porto’s national orchestra, Casa da Música, and students from the Master of Science degree in Music and Performing Arts of Oporto (ESMAE). ”
The numbers of these samples are not consistent, how do you compare?
What can these results represent?
- It is very difficult to distinguish teeth from the image obtained by using x-ray. Why not use CT?
- Overjet, Overbite and Lower facial angle will be changed due to human growth and diet patterns.
Even because of congenital diseases, everyone is different. For example, Angle Class 3 Malocclusion and Angle Class 3 Malocclusion. Therefore, how to determine the impact caused by the performance?
Discussion
- Does this study discuss how to improve the way of playing? Assuming that there will be an impact between the two, is it recommended to use another playing mode? Will this change the performance of the instrument?
- Should non-performing people be used as a control group?
Conclusions
- It should highlight the value of this research.
References
- wrong format.
Reviewer 3 Report
The aim of the study was to evaluate the tooth position of wind instrument players by comparing cephalometric values regarding dental parameters between two different  groups of musicians: wind instrument players and string players. The parameters tested were overjet, overbite, lower facial height, facial convexity, lower incisor inclination and interincisal angle.
Nowadays it is quite rare to have permission to take cephalograms only because of the study, but now the study design was approved by the ethics committee of Faculty of Dental Medicine, University of Porto, no. 880292. Thus, it was in accordance with the World Medical Association Declaration of Helsinki. To all participants a verbal explanation was given  together with a written consent explaining the objective of the study, its methods and risks/benefits.
Introduction was well written.
Material and methods:
-How many women and men were included in the study group?
-What was the age of the subjects?
-How long have they played the instrument ( it was said over 10 years)
-How many hours / per day they play? It need to have 4-6 hours / day to effect the tooth position according to literature ( Proffit et al)
-What was the molar relationship of the subjects?
-Did the subjects had also clinical examination?
-Did they had TMJ problems?
- Did they have transversal malocclusions or asymmetrical occlusion?
Results: Tables were clear. The results were interesting, but it would be good to report also if they had TMJ problems, asymmetrical occlusion or other occlusal characteristics?
Discussion was well and thoroughly written, but it was maybe too long. So it could be good to shorten it to be more pleasant to read.
There are studies concerning the TMJ problems in violin players and the effect of violin playing on the bony facial structures ( Kovero et al Acta Odont Scand 1995, 1996, EJO 1997) and it could be added to the discussion.
In line 173 -174 the authors say:This does not happen in regular patients, e.g. non musicians. Usually, when there is an increased overjet there is a reduced overbite, because there are often  parallel dysfunctional habits. How the authors explain this and do they have reference for this statement? Because patient can have and often have both increased overjet and overbite in Class II malocclusions.
In line 222, what the authors mean by lingual crossbite?
Conclusions were good and based on the results of the study.
There were a good amount of references.
I think this article is interesting.
Round 2
Reviewer 2 Report
This manuscript is well written. I recommend publication of this manuscript without any changes.
Reviewer 3 Report
Dear Authors,
Thank you for your reply and corrections you have made to the manuscript.
I think the manuscript is now worth acceptance.